# Protective Effects of *Graptopetalum paraguayense* E. Walther against Methylglyoxal-Induced Liver Damage and Microflora Imbalances Caused by High-Fructose Induction

Bao-Hong Lee [1], Siou-Ru Shen [2], Pei-Sheng Lee [2], Xin-Sen Huang [2], Wen-Chang Chang [2,*] and She-Ching Wu [2,*]

[1] Department of Horticulture, National Chiayi University, Taiwan 600355, China; bhlee@mail.ncyu.edu.tw
[2] Department of Food Sciences, National Chiayi University, Taiwan 600355, China
* Correspondence: wcchang@mail.ncyu.edu.tw (W.-C.C.); scwu@mail.ncyu.edu.tw (S.-C.W.)

**Abstract:** Methylglyoxal (MG) is a highly reactive dicarbonyl aldehyde and a major precursor of advanced glycation end products that result in oxidative stress. *Graptopetalum paraguayense* E. Walther (WGP) is a herbal medicine of Taiwan with the hepatoprotective property. The aim of this study was to investigate the protective effects of the water extract of WGP on MG-induced liver damage in a rat model. The results showed that WGP lowered the total cholesterol level and the low-density lipoprotein cholesterol level. WGP could help normalize the MG level. The amelioration of inflammatory factors such as transformation growth factor-β1 was observed in the WGP treatment group. In another animal model, a high-fructose diet (HFD) was used to induce intestinal dysfunction in C57BL/6 mice. The results indicated that the HFD induction resulted in intestinal dysbiosis, including inflammation, microflora imbalances, and reductions in tight-junction proteins. However, both WGP and its active compound gallic acid could improve intestine function. According to the above, WGP can improve hyperlipidemia in the liver, inhibit inflammatory cytokine production, and regulate intestinal flora in mice, as well as enhance the intestinal barrier. These findings provide a basis for the development of health products.

**Keywords:** methylglyoxal; *Graptopetalum paraguayense* E. Walther; high-fructose diet; inflammation; intestine; microflora

## 1. Introduction

Methylglyoxal (MG) is a highly reactive dicarbonyl aldehyde. Endogenous MG is primarily formed from dihydroxyacetone phosphate through the metabolism of fructose in glycolysis. MG is the metabolite of endogenous aminoacetone, which is oxidized by semicarbazide-sensitive amine oxidase and further catalyzed by acetol/acetone monooxygenase [1]. MG can promote glycation by interacting with DNA, lipids, and proteins, primarily with arginine, lysine, and cysteine residues, to produce advanced glycation end products (AGEs). As a main precursor of AGEs, MG plays an important role in the induction of oxidative stress [2]. In addition, MG can increase the formation of hydroxyl radicals, peroxynitrite, superoxide, hydrogen peroxide, and proinflammatory cytokines in different types of cells [3]. MG also increases oxidative stress in organisms by decreasing antioxidant levels and enzyme activities, including those of glutathione (GSH), glutathione peroxidase (GPx), glutathione reductase (GR), and manganese superoxide dismutase (SOD) [4]. The levels of MG or MG derivatives and the amount of end products of glycation in the blood were found to be higher in patients with both type I and type II diabetes [5,6]. It was reported that glucose intake was decreased in MG-treated adipocytes [7], and the apoptosis of MG-induced cells was triggered by the activation of nuclear factor (NF)-κB [8]. Another study found that insulin resistance and pancreatic damage were induced after oral feeding with MG [9].

Although the importance of gut microbiota to health is well known, it is difficult to maintain balanced gut microbiota over a long period of time due to the host's living environment, dietary habits, and health [10]. Gut microbiotas have genes that encode over 4,000,000 proteins and cause biological responses in the human body [11]. High-protein or high-fat diets will promote the ratio of the phylum *Bacteroides* in the intestine, whereas the ratio of the phylum *Prevotella* is greatly increased when large amounts of carbohydrates are consumed [12]. Another study found that a high-fat diet leads to an increase in pathogenic *Alistipes* and *Anaerotruncus*, as well as decreases in *Lactobacillus* and *Alloprevotella* in mice [13].

*Graptopetalum paraguayense* E. Walther is considered to be a folk herbal medicine in Taiwan, and it is well known for its ability to alleviate hepatic disorders. It has been reported that the freeze-dried powder of *G. paraguayense* (15 mg/mL) contains over 95% 1,1-diphenyl-2-picrylhydrazyl [14]. In the same study, the total phenolic contents of *G. paraguayense* are rich in antioxidants corresponding to 3.58 mg gallic acid equivalent/g and 1.03 mg quercetin equivalent/g [14]. An ethanol extract of *G. paraguayense* was found to inhibit T-cell infiltration, inhibit cytokine production (IL-4, IL-5, and IL-13) in Th2 cells, and regulate the immune system by activating the Nrf2 transcription factor [15]. Moreover, *G. paraguayense* was found to effectively lower insulin levels in diabetic mice by elevating peroxisome-proliferator-activated receptor-γ and pancreatic-duodenal homeobox-1. This increased insulin synthesis and inhibited the expression of CCAAT/enhancer binding protein-b [15].

In this study, we investigated the protection of a water extract of *G. paraguayense* against MG-induced liver damage and intestinal microflora imbalances caused by in vivo high-fructose induction.

## 2. Materials and Methods

### 2.1. Materials

Fresh *G. paraguayense* E. Walther was purchased from Wei-Shen farm, Chiayi, Taiwan, China. The MG solution, 2-methylquinoxaline (2-MQ), 5-methylquinoxaline (5-MQ), N-acetyl-L-cysteine (NAC), gallic acid, peroxidase, bovine serum albumin (BSA), ethylenediaminetetraacetic acid disodium salt dehydrate (EDTA-2Na), trichloroacetic acid, 5′-5′-dithio-bis (2-nitro-benzoic acid), GSH, GR, glutathione disulfide (GSSG), β-nicotinamide adenine dinucleotide (NADPH), hydrochloric acid, and acrylamide were obtained from Sigma Chemical Co. (St. Louis, MO, USA). The perchloric acid, sodium dihydrogen phosphate monohydrate ($NaH_2PO_4$), potassium phosphate dibasic ($K_2HPO_4$), and methanol were obtained from Merck (Darmstadt, Germany). The acetonitrile, tris (base), sodium dodecyl sulfate (SDS), and trishydrochloride were obtained from Sigma Chemical Co. (St. Louis, MO, USA).

### 2.2. Sample Preparation

Fresh *G. paraguayense* E. Walther was washed and minced, and then 2000 g of the minced sample was mixed with 20 L of distilled water and extracted at room temperature by ultrasonication (Ultrasonic Delta DC600H, Tainan, Taiwan, China) for 40 min. After extraction, the extracts were vacuum-freeze-dried and stored at −20 °C until use. These water extracts of *G. paraguayense* E. Walther are hereafter referred to as WGP.

### 2.3. Animal Experiments

Experiment 1: The 8-week-old male Sprague–Dawley rats were obtained from Bi-oLASCO, Taiwan Co., Ltd.(Dongshan township, Yilan country, Taiwan) All procedures regarding animal care and use were carried out in accordance with the experimental protocols, which were approved by the Institutional Animal Ethics Committee of Chiayi University, Chiayi, Taiwan (IACUC approval no. 103013). The rats were randomly divided into the following six groups (eight rats/group): (1) control, (2) MG treatment (195 mg/kg bw), (3) MG plus N-acetyl-L-cysteine treatment, (4) MG plus high dose of WGP (HWGP;

500 mg/kg bw), (5) MG plus medium dose of WGP (MWGP; 250 mg/kg bw), and (6) MG plus low dose of WGP (LWGP; 50 mg/kg bw). All of the rats were reared and maintained in a controlled environment ($23 \pm 2$ °C, $60 \pm 5\%$ relative humidity, and cyclic light of 12 h light/12 h dark). The rats were given ad libitum access to food (LabDiet@ 5001, Richmond, Commonwealth of Virginia, Richmond, VA, USA) and water, administered with MG orally twice per week, and acclimatized for 2 weeks prior to use. After 6 weeks of experimental feeding, the rats were decapitated, and their blood was collected for further evaluation. The blood was centrifuged at $1000 \times g$ for 10 min and the serum was frozen at $-20$ °C until analysis. The liver, heart, kidneys, spleen, and adipocytes of each rat were removed, rinsed with PBS, and quickly frozen with liquid nitrogen. All of the pretreated samples from the rats were stored at $-80$ °C until use.

Experiment 2: Male 5-week-old C57BL/6J mice were given a standard diet for 1 week and then randomly divided into the following six groups: (1) normal group (standard diet, NC), (2) high-fructose diet (HFD), (3) HFD plus LWGP (50 mg/kg bw), (4) HFD plus MWGP (250 mg/kg bw), (5) HFD plus HWGP (500 mg/kg bw), and (6) HFD plus gallic acid (0.5 mg/kg bw). All of the mice were reared and maintained in a controlled environment ($23 \pm 2$ °C, $60 \pm 5\%$ relative humidity, and cyclic light of 12 h light/12 h dark). The experimental animals used in this study were reviewed and approved by the Institutional Animal Care and Use Committee (IACUC) of the National Chiayi University in Taiwan, with IACUC approval no. 105042.

### 2.4. Serum Biochemical Assays

The ALT (alanine transaminase), AST (aspirate transaminase), ALP (alkaline phosphatase), TC (total cholesterol), TG (triglycerides), high-density lipoprotein cholesterol (HDL-C), and low-density lipoprotein cholesterol (LDL-C) levels in the serum samples were determined by using commercial kits obtained from Randox Laboratories Ltd. (Antrim, UK). Blood samples were taken from the venter vein of the sacrificed rat, allowed to clot for 30 min at room temperature, and then centrifuged at $3000 \times g$ for 20 min to obtain the serum, which was stored at $-80$ °C before use. Analyses were performed following the supplier's protocols.

### 2.5. Assays for Antioxidative Enzymes

The liver tissue was homogenized on ice with RIPA buffer (Cell Signaling Technology, Beverly, MA, USA) using a homogenizer and centrifuged at $12,000 \times g$ (4 °C, 60 min) to collect supernatant as the liver extract. The protein concentration in the extract was determined using Bio-Rad Protein Assay Kit (Bio-Rad, Hercules, CA, USA) with bovine serum albumin as the standard. GPx, GR, and catalase (CAT) activities analysis was performed following the supplier's protocols.

GPx activity was determined by the method previously described [5] as follows: 100 µL of homogenate was mixed with 800 µL of 100 mM potassium phosphate buffer (1 mM EDTA, 0.2 mM NADPH, 1 mM $NaN_3$, 1 unit/mL GR, and 1 mM reduced GSH) at a pH of 7.0 and incubated at room temperature for 10 min. After adding 100 µL of 2.5 mM $H_2O_2$, the enzyme reaction was initiated. The GPx activity was measured by continuously calculating the change in absorbance at 340 nm for 5 min.

For GR activity determination, 100 µL of homogenate was added to 0.1 M phosphate buffer containing 50 mM oxidized GSSG, 1 mM $MgCl_2 \cdot 6H_2O$, and 0.1 mM NADPH at a pH of 7.0. The decrease in absorbance at 340 nm was continuously calculated for 5 min.

For CAT activity determination, 50 µL of homogenate was mixed with 950 µL $H_2O_2$ (0.02 M) and incubated at room temperature for 3 min. The CAT activity was determined by continuously calculating the change in absorbance at 240 nm for 3 min [5].

### 2.6. Assays for Interleukin (IL)-6, IL-1β, IL-10, and Tumor Necrosis Factor (TNF)-α

The concentrations of IL-6, IL-10, IL-1β, and TNF-α in the serum samples were determined by using an immunoassay kit and following the manufacturer's instructions (Invitrogen Thermo Fisher Scientific, Chelmsford, MA, USA).

### 2.7. Western Blot

The proteins in the liver tissue samples were extracted with RIPA lysis buffer (phosphate-buffered saline (PBS) consisting of 1% NP-40, 1 mM phenylmethanesulfonyl fluoride, 0.1% SDS, 1 mM sodium orthovanadate, 0.5% sodium deoxycholate, and 1 mM sodium fluoride). After proteins lysis, the protein concentrations were determined as follows: 25 μg of the protein samples was mixed with an equal amount of 2X SDS-loading buffer (100 mM Tris-HCl, 20% glycerol, 4% SDS, and 0.2% bromophenol blue) for the Western blot analysis. The samples were heated at 95 °C for 5 min and electrotransferred to polyvinylidene difluoride membranes. The membranes were then blocked with tris-buffered saline with Tween 20 (TBST; 20 mM Tris-HCl, 150 mM NaCl, and 0.1% Tween 20) containing 5% BSA at room temperature for 1 h to avoid nonspecific binding. The membranes were washed with PBS twice and incubated with primary antibodies against TGF-β1, occluding, claudin-1, and glyceraldehyde-3-phosphate dehydrogenase (GAPDH) (Santa Cruz Biotechnology Inc., Santa Cruz, CA, USA) at 4 °C overnight. After extensive washing, the membranes were incubated with horseradish-peroxidase-conjugated secondary antibody in TBST at room temperature for 1 h. The proteins were detected by an enhanced chemiluminescence system (Chemi-Smart 5000 image acquisition, Vilber Lourmat, Marne-la-Vallee, France).

### 2.8. Analytical Procedure for Quantification of the MG

The MG was measured as the corresponding quinoxalines and 2-MQ, which were derived from ortho-phenylenediamine (oPD) using a reverse-phase high-performance liquid chromatography (RP-HPLC) procedure coupled with UV detection at 315 nm. Mightysil RP-18 columns (4.6 × 250 mm, 5 μm) were used for the separation of the MG. For the mobile phase, a mixture of 75% (*v/v*) 10 mmol/L $NaH_2PO_4$ (pH of 4.5) and 25% (*v/v*) acetonitrile was used for the isocratical elution. Quantification was carried out by calculating the peak areas of the samples and the internal standard with a standard calibration curve. Standard stock solutions of 2-MQ and 5-MQ were prepared with deionized water (4 mg/mL). The solutions were stored at 4 °C in the dark and used within 2 weeks.

### 2.9. Assay for Fecal Microbial Flora

Feces from the colon of each mouse were collected, immediately soaked in liquid nitrogen, and stored at −80 °C for subsequent use. The total genomic DNA from the samples was extracted using a QIAamp PowerFecal DNA Kit (Qiagen, Venlo, Netherlands). We measured the absorbance of the extracted DNA at wavelengths of 260 and 280 nm, verified the DNA concentration and purity, and stored the extracted DNA at −20 °C for later use. Amplification of the product by PCR was performed using Promega Taq DNA Polymerase (Promega Co., Madison, WI, USA). All reactions were carried out in a thermal cycler (Model 2400, Perkin-Elmer, Norwalk, CT, USA) with the following primers: *Bacteroidetes* forward: 5′-GTTTAATTCGATGATACGCGAG-3′ and reverse: 5′-TTAASCCGACACCTCACGG-3′; *Firmicutes* forward: 5′-GGAGYATGTGGTTTAATTCGAAGCA-3′ and reverse: 5′-AGCTGA CGACAACCATGCAC-3′; *Actinobacteria* forward: 5′-TGTAGCGGTGGAATGCGC-3′ and reverse: 5′-AATTAAGCCACATGCTCCGCT-3′; *Verrucomicrobia* forward: 5′-TCAKGTCAGTA TGGCCCTTAT-3′ and reverse: 5′-CAGTTTTYAGGATTTCCTCCGCC-3′; *Tenericutes* forward: 5′-ATGTGTAGCGGTAAAATGCGTAA-3′ and reverse: 5′-CMTACTTGCGTACGTAC TACT-3′; *Proteobacteria* forward: 5′-GCTAACGCATTAAGTRYCCCG-3′ and reverse: 5′-GCCATGCRGCACCTGTCT-3′; and universal forward: 5′-AAACTCAAAKGAATTGACGG-3′ and reverse: 5′-CTCACRRCACGAGCTGAC-3′ (nucleotide symbols: R = A or G; Y = C or T; N = any nucleotide; W = A or T; M = A or C; K = T or G; S = C or G; and H = A/C/T) [16]. Products of the reaction were separated on 20 g $L^{-1}$ agarose gel and stained with 1 μg/mL

ethidium bromide using a UVPGDS-7900 digital imaging system (UVP AutoChemi System, Cambridge, UK).

*2.10. Statistical Analysis*

The results were presented as means ± standard deviations (SDs). SPSS (Statistical Product and Service Solutions 22.0) was used to perform the data analysis, and one-way ANOVA was used for comparisons between groups. The post hoc Duncan's multiple range method was used for comparisons of significant differences, and $p < 0.05$ would be regarded as statistically significant.

## 3. Results and Discussion

*3.1. The Protection of WGP against Liver Damage Caused by MG Induction*

During the 6-week experiment, the body weight of the MG group rats was heavier compared to those in the MG + NAC, MG + LWGP, MG + MWGP, and MG + HWGP groups (Supplementary Figure S1). There was no significant difference in food and water intake among the groups. MG promotes glycation to produce AGEs, leading to the elevation of oxidative stress [17]. AGEs can also react covalently with several biomolecules, resulting in cellular membrane degeneration, increased permeability, and leakage of cytoplasm. Reports have revealed that the changes in AST and ALT activity can be used to evaluate cellular integrity within organisms and fibrosis [18,19]. After 4 weeks, for the group receiving the oral administration of MG, serum ALT had increased more than two-fold compared with that of the normal group (Table 1). Feeding with various doses of WGP could decrease MG-induced liver damage. HWGP decreased ALT activity by 15% compared with that in the MG treatment group. However, in terms of AST, no significant differences were observed among the different groups. ALP is mainly used to evaluate bile duct obstruction, and its activity increases when liver cells are damaged. As given in Table 1, ALP activity was increased by 47% (from 116.6 U/L to 171.4 U/L) after treatment with MG. After feeding with a high dose of WGP, ALP activity had decreased by 18% to 145.1 U/L.

**Table 1.** Effects of WGP on bioactivities of serum ALT, AST, and ALP in MG-treated rats.

| Groups [2] | ALT [3] | AST | ALP |
|---|---|---|---|
| | units/L [1] | | |
| Normal | 131.0 ± 45.9 [b] | 157.3 ± 14.2 [a] | 116.6 ± 26.7 [b] |
| MG | 281.9 ± 17.0 [a] | 209.7 ± 41.9 [a] | 171.4 ± 37.9 [a] |
| MG + NAC | 279.7 ± 32.4 [a] | 220.0 ± 88.5 [a] | 164.7 ± 16.8 [ab] |
| MG + LWGP | 235.9 ± 84.5 [a] | 176.9 ± 45.2 [a] | 125.4 ± 35.3 [ab] |
| MG + MWGP | 273.7 ± 26.9 [a] | 183.1 ± 41.8 [a] | 166.8 ± 15.3 [a] |
| MG + HWGP | 240.8 ± 26.2 [a] | 191.5 ± 25.9 [a] | 145.1 ± 16.2 [ab] |

[1] Each value is expressed as the mean ± SD ($n = 6$). [a,b] Values in a column with different superscripts are significantly different ($p < 0.05$). [2] MG, 195 mg/kg bw; NAC, 100 mg/kg bw; LWGP, low-dose WGP (50 mg/kg bw); MWGP, medium-dose WGP (250 mg/kg bw); HWGP, high-dose WGP (500 mg/kg bw). [3] ALT, alanine aminotransferase; AST, aspartate aminotransferase; ALP, alkaline phosphatase; WGP, water extract of *G. paraguayense* E. Walther; MG, methylglyoxal; NAC, *N*-acetyl-L-cysteine.

*3.2. Serum Biochemical Values*

Treatment with MG could lead to significant elevations of serum TC and TG levels compared with the normal group (Table 2). The oral administration of NAC and WGP could decrease the levels of both TC and TG, and, in particular, the high dose of WGP had the most robust effect among them. As given in Table 2, NAC and WGP in varying doses decreased serum LDL-C levels from 76.5 ± 4.9 mg/dL (MG-treated group) to 50.1 ± 13.9, 47.2 ± 21.6, 52.9 ± 15.5, and 50.1 ± 15.8 mg/dL (MG plus NAC, MG plus LWGP, MG plus MWGP, and MG plus HWGP groups, respectively). It has been reported that people with LDL-C levels over 130 mg/dL or LDL-C/HDL-C ratios over 3.5 are confronted with higher risks for cardiovascular disease [20]. Table 2 shows that the LDL-C/HDL-C ratio

increased to 3.44 after the rats were treated with MG, whereas the ratio decreased after WGP treatment. A previous report has shown that MG adducts are correlated with serum LDL-C and TG [21]. The clinical trial result showed that blood pressure, fasting glucose, and LDL-C levels were significantly lower and HDL-C level and antioxidant enzymes activities were significantly higher after WGP supplementation [22]. WGP has the ability to break down lipid peroxides present in cells and lipoproteins circulating in the body, thereby inhibiting LDL oxidation. This action helps to prevent the formation of ox-LDL, which is the initial stage in the development of atherosclerotic plaques [23]. This study indicated that treatment with WGP has protective effects and reduced lipid metabolic disorders in the liver.

**Table 2.** Effects of WGP on serum TG, TC, HDL-C, and LDL-C in MG-treated rats.

| Groups [2] | TG [3] | TC | HDL-C | LDL-C | LDL/HDL |
|---|---|---|---|---|---|
| | mg/dL [1] | | | | |
| Normal | $16.4 \pm 3.1$ [ab] | $69.3 \pm 4.2$ [b] | $17.6 \pm 2.1$ [a] | $49.6 \pm 3.4$ [b] | $2.73 \pm 0.4$ [a] |
| MG | $19.9 \pm 2.7$ [a] | $102.8 \pm 2.6$ [a] | $22.4 \pm 2.8$ [a] | $76.5 \pm 4.9$ [a] | $3.44 \pm 0.6$ [b] |
| MG + NAC | $20.8 \pm 5.6$ [a] | $79.1 \pm 18.6$ [ab] | $24.8 \pm 6.6$ [a] | $50.1 \pm 13.9$ [b] | $2.07 \pm 0.6$ [a] |
| MG + LWGP | $18.4 \pm 11.4$ [ab] | $84.2 \pm 7.3$ [ab] | $33.3 \pm 18.6$ [a] | $47.2 \pm 21.6$ [b] | $2.02 \pm 1.6$ [a] |
| MG + MWGP | $12.6 \pm 3.7$ [ab] | $81.5 \pm 21.7$ [ab] | $26.1 \pm 5.7$ [a] | $52.9 \pm 15.5$ [b] | $2.01 \pm 0.2$ [a] |
| MG + HWGP | $10.1 \pm 3.2$ [b] | $79.8 \pm 25.5$ [ab] | $24.2 \pm 5.4$ [a] | $50.1 \pm 15.8$ [b] | $1.87 \pm 0.6$ [a] |

[1] Each value is expressed as the mean $\pm$ SD ($n$ = 6). [a,b] Values in a column with different superscripts are significantly different ($p < 0.05$). [2] MG, 195 mg/kg bw; NAC, 100 mg/kg bw; LWGP, low-dose WGP (50 mg/kg bw); MWGP, medium-dose WGP (250 mg/kg bw); HWGP, high-dose WGP (500 mg/kg bw). [3] ALT, alanine aminotransferase; AST, aspartate aminotransferase; ALP, alkaline phosphatase; WGP, water extract of *G. paraguayense* E. Walther; MG, methylglyoxal; NAC, *N*-acetyl-L-cysteine.

### 3.3. Effects of WGP on Hepatic Antioxidative Enzymes

Antioxidative enzymes are recognized to be frontline protection against oxidative damage in biological molecules. It has been shown that MG and its derivatives can decrease antioxidants and antioxidative enzymes, leading to increased oxidative stress [15,17,19]. As given in Table 3, compared with the normal group, the activities of the antioxidative enzymes GR, GPx, and CAT in the rats receiving MG treatment were decreased by 30.2%, 23.3%, and 42.7%, respectively.

**Table 3.** Effects of WGP on the activities of antioxidative enzymes in the livers of MG-treated rats.

| Groups [2] | GR [3] | GPx | CAT |
|---|---|---|---|
| | nmol NADPH/min/g of Protein [1] | | $\mu$mol $H_2O_2$/min/g of Protein |
| Normal | $204.4 \pm 38.0$ [a] | $144.4 \pm 1.8$ [bc] | $605.2 \pm 36.1$ [ab] |
| MG | $142.6 \pm 8.3$ [bc] | $110.7 \pm 11.2$ [d] | $346.8 \pm 20.5$ [c] |
| MG + NAC | $130.4 \pm 25.0$ [c] | $143.9 \pm 6.3$ [bc] | $549.8 \pm 37.2$ [b] |
| MG + LWGP | $128.5 \pm 5.9$ [c] | $136.1 \pm 11.8$ [cd] | $643.0 \pm 72.4$ [ab] |
| MG + MWGP | $185.4 \pm 31.9$ [ab] | $175.8 \pm 21.8$ [a] | $672.9 \pm 63.2$ [a] |
| MG + HWGP | $204.5 \pm 37.2$ [a] | $171.2 \pm 25.6$ [ab] | $685.3 \pm 52.1$ [a] |

[1] Each value is expressed as the mean $\pm$ SD ($n$ = 6). [a–d] Values in a column with different superscripts are significantly different ($p < 0.05$). [2] MG, 195 mg/kg bw; NAC, 100 mg/kg bw; LWGP, low-dose WGP (50 mg/kg bw); MWGP, medium-dose WGP (250 mg/kg bw); HWGP, high-dose WGP (500 mg/kg bw). [3] ALT, alanine aminotransferase; AST, aspartate aminotransferase; ALP, alkaline phosphatase; WGP, water extract of *G. paraguayense* E. Walther; MG, methylglyoxal; NAC, *N*-acetyl-L-cysteine.

The HWGP treatment increased GR activity by 30.0% to 204.5 nmol/min/g of protein. These antioxidative effects were dose-dependent. After feeding with LWGP, MWGP, and HWGP, the GPx activities were determined to be 136.1, 175.8, and 171.2 nmol/min/g of protein, respectively. The results indicated that LWGP, MWGP, and HWGP elevated GPx activity by 2.9%, 58.8%, and 54.7%, respectively, compared to the MG treatment group. MG induced decreases in CAT activity, which was increased to 672.9 and 685.3 µmol/min/g of

protein (11.2% and 13.2%, respectively) after MWGP and HWGP treatments. The study showed that WGP increased hepatocyte cellular GSH level and antioxidant enzymes, including superoxide dismutase, glutathione reductase, and catalase in $CCl_4$-induced rats [24]. The clinical trial results showed that antioxidant enzyme activities were significantly higher after WGP supplementation for 12 weeks in humans [22].

### 3.4. WGP Suppressed MG-Induced Inflammatory Cytokines

TNF-α and IL-6 play important roles in the process of liver damage and are usually used to evaluate the extent of inflammation within organisms. IL-10 and TGF-β1 are anti-inflammatory cytokines that can inhibit the formation of proinflammatory cytokines. The level of TGF-β1 is currently one of the indices for observing liver fibrosis [8,9]. The TNF-α levels of the MG group were found to be higher than those of the other groups receiving varying doses of WGP (Figure 1A). Compared with the normal group (2.80 pg/mg of protein), the TNF-α level increased 2.31-fold to 6.48 pg/mg of protein after the MG treatment. The previous study showed that the group of rats treated with $CCl_4$ along with WGP (150 and 300 mg/kg bw) demonstrated a decrease in TNF-α levels in their serum [24].

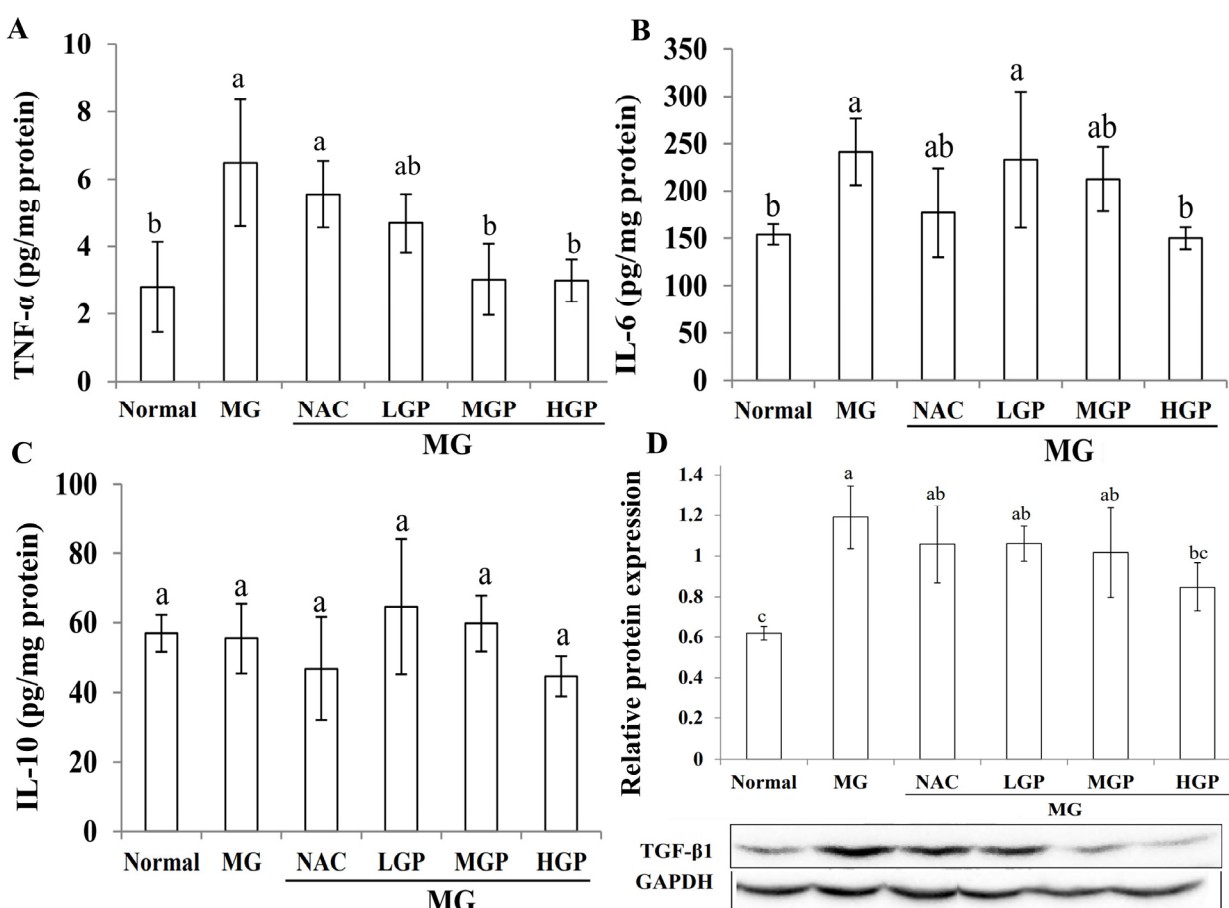

**Figure 1.** Effects of WGP on cytokines such as (**A**) TNF-α, (**B**) IL-6, (**C**) IL-10 (ELISA measurement), and (**D**) TGF-β (Western blot) contents in the livers of MG-treated rats. Each value is expressed as the mean ± SD ($n = 6$). [a–c] Values in a column with different superscripts are significantly different ($p < 0.05$). MG, 195 mg/kg bw; NAC, 100 mg/kg bw; LWGP, low-dose WGP (50 mg/kg bw); MWGP, medium-dose WGP (250 mg/kg bw); HWGP, high-dose WGP (500 mg/kg bw); WGP, water extract of *G. paraguayense* E. Walther; MG, methylglyoxal; NAC, *N*-acetyl-L-cysteine.

The inhibition rates of the NAC group and the groups receiving different doses of WGP (LWGP, MWGP, and HWGP) were 14.4%, 27.6%, 53.4%, and 53.7%, respectively. In the MG treatment group, IL-6 was increased 1.57-fold compared with the normal group,

and this level decreased after treatment with HWGP (Figure 1B). There was a significant reduction in levels of inflammatory markers in the group that received WGP as compared to the placebo group [25]. The levels of IL-10 in the rat livers were not significantly different among the groups (Figure 1C). The relative protein expression level in the MG group was 1.19, which was 1.9-fold greater than that of the normal group. The results indicated that MG could induce liver fibrosis in rats. After six experimental periods, the relative TGF-β1 expression levels in the rat livers decreased after treatment with HWGP (Figure 1D). The study showed that WGP extracts inhibited the proliferation and migration via suppression of the TGF-β1 pathway in rat hepatic stellate HSC-T6 cells [26].

### 3.5. HPLC Analyses of the MG Concentrations in the Rats

MG concentration is strongly related to diabetes [4,5] and the formation of highly reactive glycation products. Currently, the most common measurement method for MG is to quantify 2-MQ, which is the product of the interaction between o-PD and MG. The 2-MQ isomer 5-MQ was used as the internal standard to increase the accuracy of detection (Figure 2A). The influence of WGP on the serum MG levels of the rats receiving MG induction is shown in Figure 2B. The serum MG levels in the rats with MG-induced liver damage were higher than those of the normal group (Figures 2B and 3A). After feeding with varying doses of WGP, the MG levels in the serum samples were significantly decreased. In the MG treatment group, the MG levels in the livers were also significantly increased—1.73-fold that of the normal group (Figures 2C and 3B). The MG levels were not significantly different among the sample treatment groups (LWGP, MWGP, and HWGP). These results indicated that the beneficial effects of WGP could be achieved even at a low dose (50 mg/kg/bw).

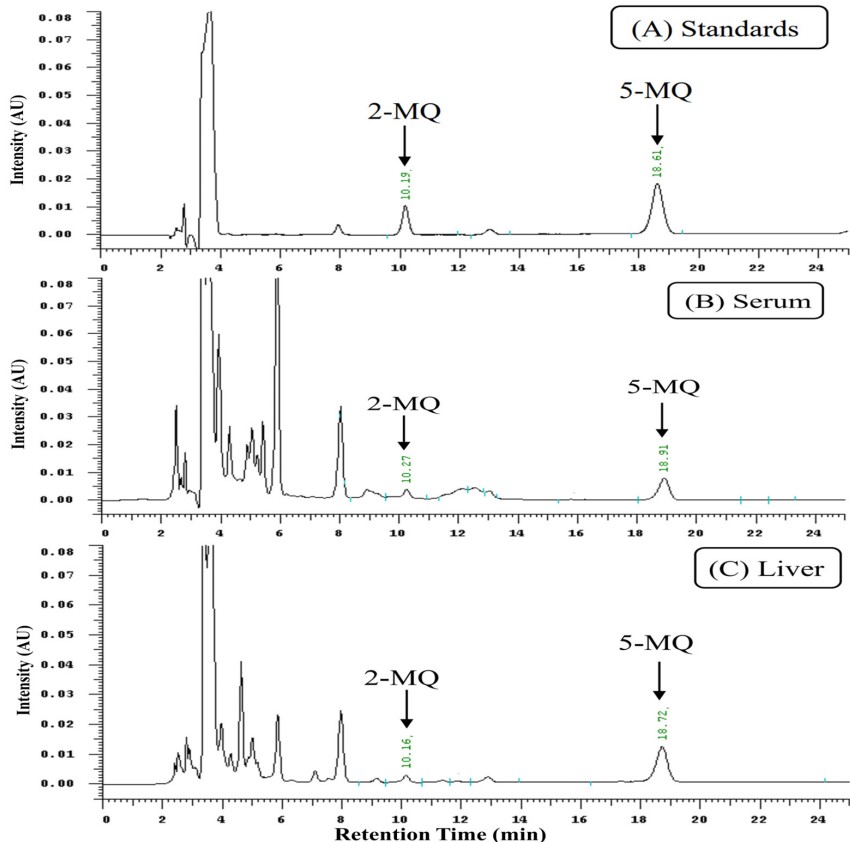

**Figure 2.** HPLC chromatograms obtained from detecting 2-MQ and 5-MQ. The peaks in (**A**) correspond to the standards, those in (**B**) are from the rats' serum samples, and those in (**C**) are from the livers of MG-treated rats. HPLC, high-performance liquid chromatography; MQ, methylquinoxaline.

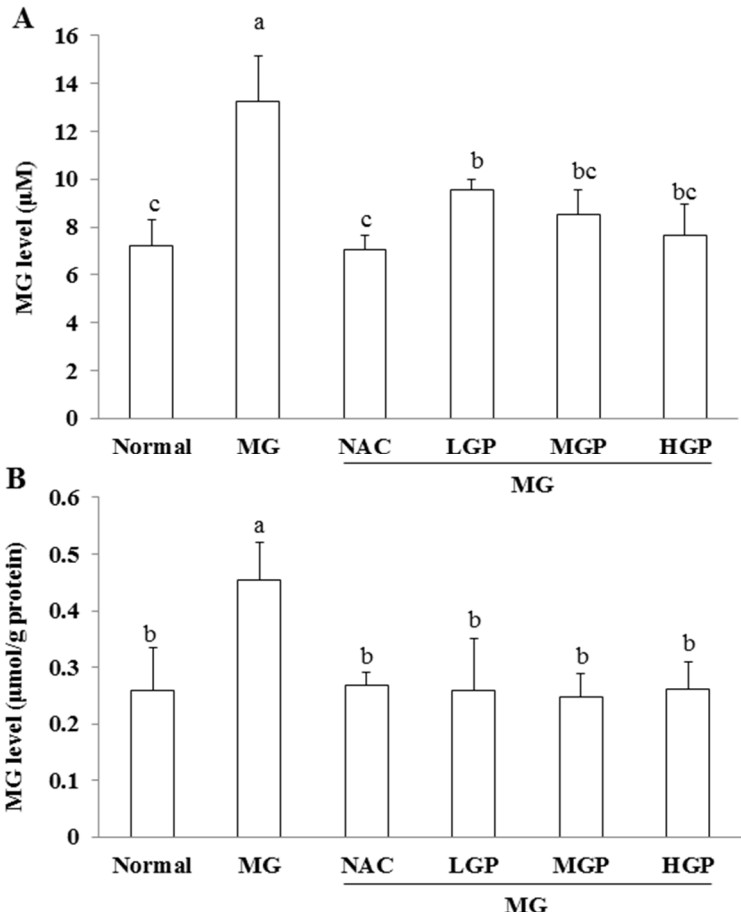

**Figure 3.** Effects of WGP on methylglyoxal levels: (**A**) the results from the serum samples and (**B**) the results from the livers of MG-treated rats. Each value is expressed as the mean $\pm$ SD ($n = 6$). [a–c] Values in a column with different superscripts are significantly different ($p < 0.05$).

MG and AGEs are both highly reactive electrophilic dicarbonyl aldehyde compounds formed during glycolysis and generated endogenously through lipid peroxidation and glucose oxidation [4,5]. MG can increase ROS formation, DNA oxidation, and protein carbonylation, and it can also decrease the mitochondrial membrane potential in isolated rat hepatocytes [8–10,27]. MG can be metabolized to the form of AGEs, which increase the risk of developing inflammatory diseases [9].

Several studies have indicated the pathological roles of MG in the liver [19]. In a previous study, animals that were exposed to MG showed significant changes in redox homeostasis in their livers [28]. Seo et al. confirmed MG-induced hepatic oxidative damage by measuring the levels of plasma ALT and AST in mice. HDL-C was shown to scavenge cholesterol in peripheral blood vessels and transfer it to the liver for metabolization [28]. Higher LDL-C concentrations can promote cholesterol accumulation on blood vessel walls and interactions with fibrinogen, blood platelets, and macrophages to cause the formation of atherosclerotic plaques which could block blood vessels [29]. We found that WGP treatment could improve and lower the high serum concentration levels of ALT, ALP, TG, and LDL-C that were caused by glycation precursors. WGP can lessen liver damage likely due to its potent effects on decreasing free radical production and increasing the activities of antioxidative enzymes. The increased activities of AST, ALT, and ALP in serum are primarily because of the leakage of these enzymes from the liver cytosol into the bloodstream [30]. We hypothesized that WGP may protect the liver against MG-induced damage by modulating free radical scavenging, antioxidative enzyme stimulation, and inflammatory cytokine inhibition.

### 3.6. The Effect of WGP on Intestinal Function and Microbiota

It has been reported that the symptoms of inflammatory bowel disease in animals are increased colonic weight and shortened mucosal length [10]. Figure 4 shows the effect of WGP on the lengths of the large intestines of mice fed an HFD. The results showed that the average length of the large intestines of the mice in the control group was 85.50 mm, and the average length of the large intestines of the mice in the HFD group was 70.25 mm. There was a significant difference between these groups. The average length of large intestines of mice treated with different doses of WGP showed an increasing trend, and the average lengths were 75.0, 75.67, and 77.00 mm for the LWGP, MWGP, and HWGP groups, respectively. Moreover, the average length of large intestines after gallic acid treatment was 76.25 mm. These results showed that WGP could alleviate the shortening of large intestine tissue caused by intestinal inflammation caused by HFD induction, and they also showed which active ingredients contained in the extract may be phenolic compounds, such as gallic acid [14].

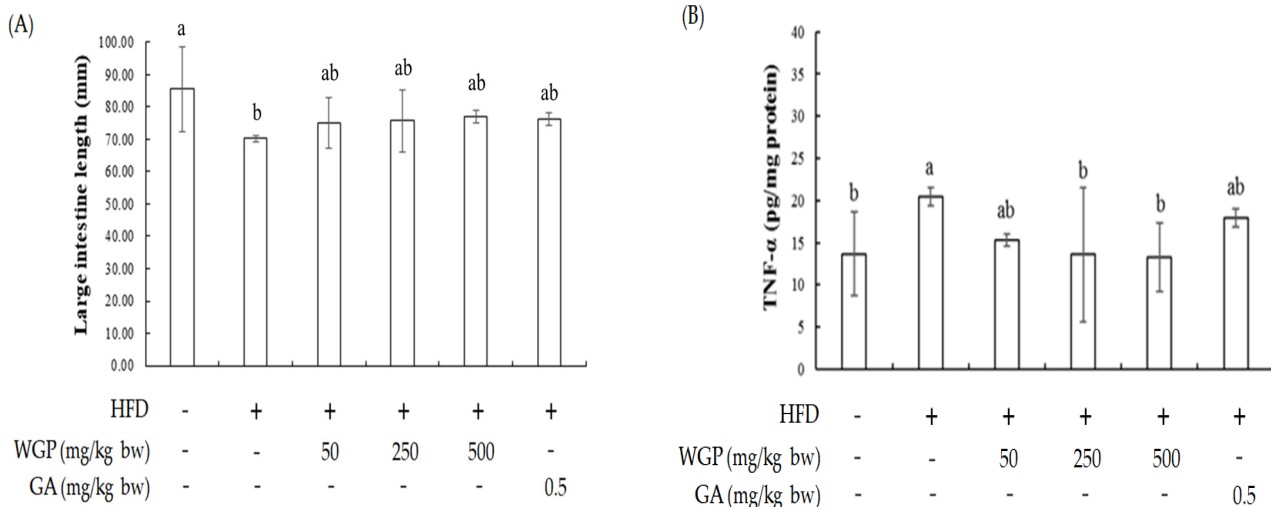

**Figure 4.** (**A**) Large intestine lengths. (**B**) TNF−α levels in the HFD−induced C57BL/6J mice treated with WGP. The values shown are the means ± SDs (*n* = 6). Superscript letters in the same column ([a] and [b]) denote significant differences ($p < 0.05$). GA, gallic acid.

On the other hand, the elevation of intestinal TNF-α levels caused by HFD induction was reduced in the WGP-treated groups. Previous studies have shown that proinflammatory cytokines, including TNF-α and IL-6, cause an increase in intestinal permeability by reducing tight proteins' expression and contributing to the inflammatory process. Altered microbial composition, which is termed dysbiosis, has been implicated in mucosal barrier dysfunction and inflammatory responses [31,32].

Fructose has been shown to affect the composition of the intestinal flora, thereby resulting in imbalances in the intestinal flora, and it reduces the levels of Bacteroidetes but elevates the levels of Firmicutes. The ratio of Bacteroidetes to Firmicutes is often used as an indicator for assessing health. Firmicutes promote body fat accumulation and are highly correlated with obesity. The interaction between the composition of intestinal flora and the gut–brain may be the initial factor that promotes the development of obesity pathology [33,34]. As shown in Figure 5, the relative content of Bacteroidetes in the control group was higher than the Firmicutes content. After being treated with HFD, the content of Firmicutes tended to increase while the relative contents of Actinobacteria and Tenericutes decreased. After treatment with different doses of WGP, the relative content of Firmicutes decreased and the ratio of Bacteroidetes to Firmicutes increased, and the content of Bacteroidetes increased significantly in the HWGP and gallic acid groups while the ratio of Bacteroidetes to Firmicutes returned to a similar ratio as that of the control group.

However, the relative content of Proteobacteria showed an upward trend after treatment with WGP and gallic acid, which was similar to the relative content of Proteobacteria in the HFD group.

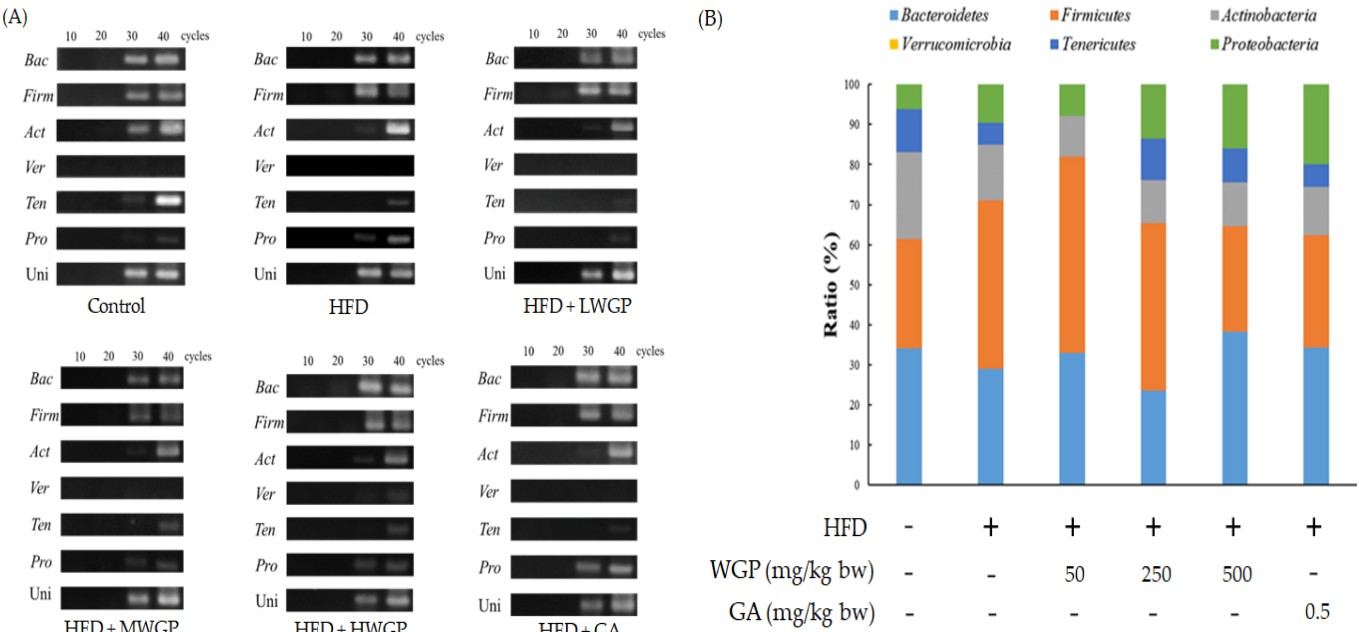

**Figure 5.** Effects of WGP on microbial composition in the feces of the HFD−induced C57BL/6J mice: (**A**) PCR measurements and (**B**) percentages. *Bac, Bacteroidetes; Firm, Firmicutes; Act, Actinobacteria; Ver, Verrucomicrobia; Ten, Tenericutes; Pro, Proteobacteria;* Uni, universal.

Figure 5A shows the percentages of microorganisms in each group under 30 cycles of PCR. The results showed that the proportion of Bacteroidetes in the control group was 34.12%, the proportion of Firmicutes was 27.29%, the proportion of Actinobacteria was 21.65%, and the proportion of Tenericutes was 10.82%. However, in the HFD group, the proportion of Bacteroidetes was 28.90% and the population of Firmicutes rose to 42.04%, but the ratio of Tenericutes fell to 5.54%. The HFD-induced mice were treated with HWGP and GA, and the relative percentages of Bacteroidetes were 38.24% and 34.36%, respectively. The Firmicutes proportions in these groups decreased to 26.37% and 27.92%, respectively, and these results were similar to those of the control group. For Tenericutes, the populations were 11.08% and 5.54% in the control and HFD groups, respectively. However, the levels of Tenericutes increased in the MWGP and HWGP groups.

Proteobacteria levels increased to 7.94%, 13.50%, and 16.04% after treatment with different doses of WGP from 6.86% (NC group) and 9.52% (HF group), and the Proteobacteria level in the GA group was 20.13%. Huang et al. (2016) reported that a high-fat diet induced an imbalance in the intestinal flora of rats. After treatment with the phenolic compounds quercetin and catechin, the relative abundance of Proteobacteria in the intestinal flora of the rats fed a high-fat diet was increased [35]. From these results, it was speculated that some bacterial species in Proteobacteria increased due to the presence of the phenolic compounds.

In many studies, the ratio of the relative abundance of Bacteroidetes and Firmicutes is often used as an indicator of intestinal flora imbalances [36]. Recent research results have shown that Proteobacteria can be used as an indicator of imbalanced intestinal flora microbes, and the relative abundance of Proteobacteria in gastric bypass, metabolic syndrome, and cancer patients is higher than that of normal healthy adults while their overall gut flora diversity is also lower [37]. WGP may promote the proliferation of some bacteria in Proteobacteria so that the overall relative content increases. However, the Bacteroidetes/Firmicutes ratio also tends to increase after treatment with WGP, indicating that WGP stabilizes the balance of intestinal flora.

The intestinal barrier is a physiological barrier that prevents the invasion of intestinal microorganisms and microbial products. The adherens junctions and desmosomes are substances that mainly connect to the intestinal epithelial cells and form a selectively permeable intestinal epithelial barrier. It has been pointed out that the long-term intake of fructose will cause a reduction in tight-junction proteins, such as occludin and claudin-1 [38]. In the results shown in Figure 6, the levels of occludin and claudin-1 were significantly decreased in the mice treated with the HFD compared to the control group. It is also shown that HFD reduced the expression of tight-junction proteins in the intestinal barrier of the mice, and it increased intestinal permeability. However, the treatments with different doses of WGP resulted in 2.60-, 2.75-, and 2.08-fold increases in occludin. In addition, claudin-1 was also increased by different doses of WGP to 3.91- (LWGP), 3.96- (MWGP), and 8.35-fold (HWGP) compared to the HFD-induced group.

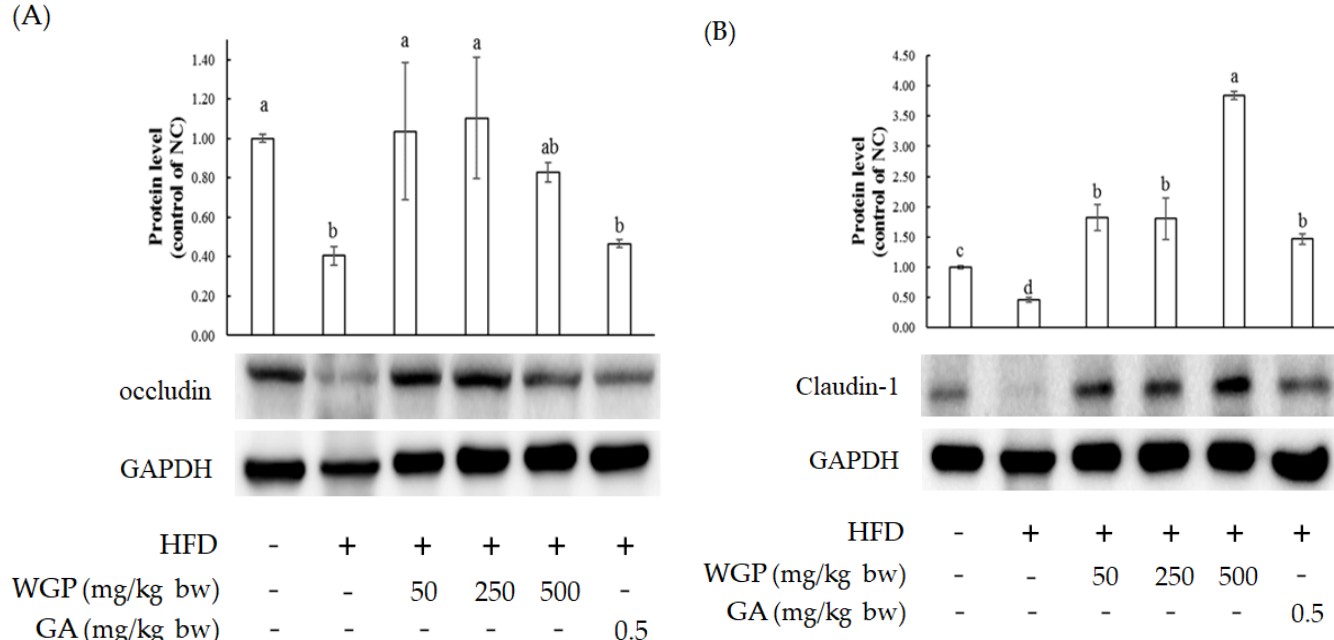

**Figure 6.** Effects of WGP on tight−junction proteins, including (**A**) occludin and (**B**) claudin−1, in the large intestines of the HFD−induced C57BL/6 mice. The values are shown as the means ± SDs (n = 6). The different superscript letters in the same column ([a–d]) denote significant differences ($p < 0.05$).

The study by Shigeshiro et al. (2013) showed that the density of tight-junction proteins, such as occludin, JAM-A, and claudin-3, decreased in a model of DSS-induced colonic inflammation in mice, but different phenolic compounds, including quercetin, curcumin, hesperetin, and naringenin, could promote tight-junction proteins in the colons of mice [39]. After treatment with (-)-Epicatechin, the expression levels of the tight-junction proteins, occludin, claudin-1, and ZO-1, in the mice had increased [40].

## 4. Conclusions

In summary, our findings showed that WGP enhanced antioxidant activities and protected against MG-induced inflammatory liver damage in rats and against HFD-induced intestinal dysbiosis in mice. WGP can improve hyperlipidemia in the liver, inhibit inflammatory cytokine production, and regulate intestinal flora in mice, as well as enhance the intestinal barrier. The results confirm that WGP has hepato- and intestinal-protective potential for being developed as a functional food.

**Supplementary Materials:** The following supporting information can be downloaded at: https://www.mdpi.com/article/10.3390/fermentation9040366/s1. Figure S1. Changes in body weight of MG-induced rats administrated with WGP for six weeks. Each value is expressed as the mean ($n$ = 6).

**Author Contributions:** B.-H.L. and W.-C.C. designed the framework of the study and analyzed the data; S.-R.S. and P.-S.L. performed most of the experimental assays; X.-S.H. and W.-C.C. revised the manuscript; P.-S.L. and S.-C.W. participated in the study design and provided certain scientific suggestions and draft corrections; W.-C.C. and S.-C.W. were responsible for the financial resources and funds for the project, supervision of the research activities, and manuscript submission. All authors have read and agreed to the published version of the manuscript.

**Funding:** The authors would like to thank the Ministry of Science and Technology of the Republic of China (ROC), Taiwan, for financially supporting this research under contract no. MOST 111-2221-E-415-001.

**Institutional Review Board Statement:** The experimental animals used were reviewed and approved by the Institutional Animal Care and Use Committee (IACUC) of the National Chiayi University in Taiwan, with IACUC approval no. 103013 (date of approval: 31 July 2017) and 105042 (date of approval: 31 July 2018) for this study involving animals.

**Informed Consent Statement:** Not applicable.

**Data Availability Statement:** The data that support the findings of this study are available in the supplementary material of this article.

**Acknowledgments:** The authors would like to thank the Ministry of Science and Technology of the Republic of China (ROC), Taiwan, for financially supporting this research.

**Conflicts of Interest:** The authors declare no conflict of interest.

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
