# Peer review of "Protective Effects of Graptopetalum paraguayense E. Walther against Methylglyoxal-Induced Liver Damage and Microflora Imbalances Caused by High-Fructose Induction"

_fermentation, doi:10.3390/fermentation9040366_

Round 1

Reviewer 1 Report

This is an interesting article trying to investigate the effects of Graptopetalum paraguayense E. Walther on Methylglyoxal induced liver damage and on HFD diet mice. The methodology is well structured and presented and the results are really interesting. One concern is the English that need substantial revision. Also, in the introduction I think you should rethink the aim of this study. If your aim was to investigate the effects of Graptopetalum in MD and HFD, then you may introduce Graptopetalum and its actions more in the Introduction instead of MD. Starting with MD seems like your main aim was MD’s actions, although you focus on both MD and HFD experiments.

Some minor comments:

-line 11 that result

--line 11 Graptopetalum paraguayense E. Walther is a herbal medicine of Taiwan with hepatoprotective

-line 21 and its active compounds

-line 21 improve intestine function

-23 regulate

-line 24 Thess results serve as the basis for the development of health products please rephrase

-line 35 AGEs not AGE

-LINE 36 IN addition

-LINE 45 were induced

-47 in health

-line 51 you mean promote ratio or abundance? As well as of Prevotella?

-line 57 is well known

-line 62 regulates

-line 67 In this study we investigated

-line 91 add ethics approval number

-91 use were carried out? What do you mean, please rephrase

-explain why you use N acetylcysteine, and why you do not use acetylcysteine but gallic in the other experiment, you should explain in detail everything you do in your experiments

-line 119 more details on the kits you have used, exact name and more details on the treatment of tissues, homogenization? How?

-line 122, 134 you should put references with numbers

-in 2.5 you do not describe in which biological samples you measured antioxidant activity, serum, tissue, both, homogenisation?

-line 138 what is MT100? Please be more specific on the kits

-2.7 Western. Again describe homogenization, the use of GADPH? As a housekeeping? Please be as specific as you can for readers without experience on the field.

-2.9 again specify in which samples

-line 173 please rephrase

-line 200 which lead

-Results. Tables. Please clarify what superscripts compare, same superscripts statistically significant difference? For example in Table 1 in AST what does it mean that all superscripts are the same? It is not clear where there is a difference or not. Please correct or be more specific in all tables and figures.

-line 223 could decrease or decreased?

-Table 2, zero is not necessary before ie 79,1

-line 227, please rewrite your discussion, you should talk about the role of LDL and TG in cardiovascular disease in general and not compare data from human with rats

-line 230. The report showed?

-line 232 reduced lipid metabolic disorders?

-line 248 had a manner

-line 252 please rephrase, not clear what you mean

-3.3 and 3.4 please comment more on your findings and compare with other studies

-3.4 cytokine production (specify in which biological samples)

-line 289 MG levels were not significantly different among the 288 sample treatment groups (LWGP, MWGP and HWGP) but lower than MG? again superscripts in figures not clear

-line 295 MG can be metabolized to the form of AGEs, which increase the risk of developing inflammatory diseases

-Line 329 please rephrase. Also define how you measure length of intestine and how you measured TNF, where?

-345 gallic?

-line 352 by reducing the levels of Bacteroidetes and increasing the levels of Firmicutes?

-line 350 tended not was tended, what do you mean by tended?

-line357-366 please rephrase

-387, regarding proteobacteria they were increase to 7,94% from???

-394 in many studies

-395rewrite

-line 402 stabilizes

-line 403 entry where?

;line 408 omit was

-line 412 please explain why occluding decreases and then increases with higher dose

-The conclusion needs rewriting, correct English and include more conclusions and suggestions

Reviewer 2 Report

This paper studied the protective effects of Graptopetalum paraguayense E. Walther 2 extract against methylglyoxal-induced liver damage and high-fat diet-induced microflora imbalance. This is very informative research. The following are some minor questions that need to be addressed.

1.      L36, change In additional to In addition.

2.      Check the temperature unit.

3.      Section 2.2 add the extraction temperature.

4.      Add the basic diet information of rats in Section 2.3.

5.      Figure 5, what are 10,20,30,40 represent? PCR cycles? It is better to redraw Figure 5A with only 40 cycles data and combine different data groups for better comparison.

6.      Add the weight of rats to the results.

7.      In the figure and table captions, please provide information about the model of the rats, MGO-treated or HFD-treated rats.
